# Robust Touch Screen Readout System to Display Noise Using Multireference Differential Sensing Scheme for Flexible AMOLED Display

**DOI:** 10.3390/mi13060942

**Published:** 2022-06-14

**Authors:** Junmin Lee, Hyoyoung Kim, Juwon Ham, Seunghoon Ko

**Affiliations:** Department of Electronic Materials Engineering, Kwangwoon University, Seoul 01897, Korea; junmin8950@gmail.com (J.L.); rlagydud07201@naver.com (H.K.); zoowon821@kw.ac.kr (J.H.)

**Keywords:** mutual capacitance, differential sensing of TSP, discrete-time analog sampling

## Abstract

This paper presents a front-end architecture for touch screen panel (TSP) readout in a TSP-integrated, ultrathin flexible display to mitigate severe display noise interference, which is an uncommon mode caused by the large panel load of the TSP in the flexible display. The differential sensing method with multireference TSP channels minimized an imbalance of the phase and amplitude of the coupled-display noise interference. In addition, cascaded time-discrete bandpass sampling was employed to enhance the touch sensitivity in the sensing block. Moreover, a rated front-end block could be reconfigured to a differential or single-ended sensing structure, which reused the prefilter capacitors in the differential sensing for offset cancellation in reference capacitance sensing. To further improve the sensitivity, programmable postfiltering was employed on the reference TSP channels. Subsequently, the proposed front-end was implemented in a 350 nm process, wherein it achieved a SNR of 50.5 dB with a scan rate of 200 Hz and attenuated aggravated display noise interference by more than 6.84 dB as compared to the conventional differential sensing method. The designed chip occupied an area of 4.8 mm^2^ and consumed 17.6 mW from a 3 V supply.

## 1. Introduction

Recently, the increasingly competitive mobile device markets and the high demand for a new form factor have enabled the commercialization of foldable mobile devices [1]. Based on this trend, the design of flexible active-matrix organic light-emitting diode (AMOLED) displays has garnered considerable interest owing to their high optical transparency, thinness, and light-weight characteristics. As the first step towards the implementation of a foldable AMOLED, a thin-film transistor (TFT) backplane was mounted on a flexible substrate material. A polyimide [2] substrate is considered the most suitable candidate in such emerging technology, because it exhibits conventional glass-like properties and can withstand bending radii smaller than 5 mm without mechanical breakdown. 

The encapsulation layer laminating process is vital for the bending characteristic, which protects the TFT from oxidation and moisture permeability. In particular, thin-film encapsulation (TFE) is employed for rigid AMOLEDs instead of glass encapsulation, because it can reduce the bending radius and relieve the stress caused by folding. For truly inward and outward bending, the TFE requires to be thinner than 10 µm [3,4]. In the transition from rigid to flexible AMOLED structures, the TSP readout system becomes more vulnerable to noise interference. Owing to the reduced thickness of the TFE in the flexible AMOLED, the display noise is largely coupled through the parasitic capacitors of the TFT, cathode, and TSP in the TSP electrodes during a display frame update [5,6]. 

These conditions become more severe as the voltage of the pixel-update signal alters from high to low, or vice versa, during every horizontal-line update. Generally, this phenomenon occurs when the display driver IC (DDI) updates the horizontal zebra pattern, which contains an alternating block and a white color line [7]. In addition to the reduced panel thickness, the metal-mesh TSP sensor replaces the indium tin oxide (ITO) sensor in the flexible display, as it can be flexible and increase the conductivity because of the reduced resistance. However, the placement of metal-mesh sensors over the organic cells of RGB is prohibited because of the Moire effect or the interference of light occurring between the metal-mesh sensor and organic cell that degrades the visibility. Thus, the TSP sensor in the flexible AMOLED comprises numerous fine-line metal mesh lines with large openings over the organic cells, which can vary the capacitance only to tens of femtofarads and the electrode resistance of the TSP up to tens of kilo-ohms upon a finger touch.

Several differential front-end architectures have been proposed to negate the common-mode interference [8,9,10,11]. In particular, a column-parallel capacitance difference (∆C_m_) in the adjacent receiving electrodes is detected by a fully or pseudo-differential readout circuit. They detect and directly process the ∆C_m_ formed between adjacent RX electrodes. However, there has been no consideration of general use environment. A common user scenario such as multi-finger gaming is described in Figure 1a, where two fingers with large touched area were placed across the whole TSP region. In a single-ended sensing, capacitance change (ΔC_F_) by a finger was sensed correctly, whereas the capacitance difference upon the finger-touches (ΔC_F_s) can be recognized as a common noise and ignored in the differential sensing. A constant can be used as an integral basis for capacitance reconstruction. However, the capacitance on TSP is largely fluctuated by temperature variation of mobile devices, making a significant offset between the constant and measured reference C_m_ [12]. Subsequently, it should be reconstituted into a single-ended C_m_ with a capacitance integral in the horizontal direction based on time-varying reference capacitance. A conceptual diagram of the differential TSP sensing in a flexible AMOLED is illustrated in Figure 1b. The TSP driver transmits an excitation signal to the transmitter electrode, and C_m(m,n)_ is formed at the cross-section of the transmitter (TX[m]) and receiver (RX[n]) electrodes. Moreover, the AMOLED pixels below the TSP electrodes are refreshed every T_sync_ (= 1/f_sync_). This is synchronous to scan signals (Scan[H:1]), whose frequency is calculated as the product of the display refresh rate and the horizontal resolution (H) of the display. Accordingly, two criteria should be satisfied to remove the common noise in the differential sensing architecture. The first criterion is to set a clean reference C_m_ in an RX electrode. For the simplicity of explanation, let us consider the C_m_s at RX[1] electrode as the reference C_m_s. Thus, the reconstructed C_m_rct(m,n)_ of the RX[n], which is an integral of ∆C_m(m,k)_, can be expressed as
(1)$$\;\;\;\;\;\;\;\;\;C_{m\_rct\left(m,n\right)}=\sum_{k=1}^{n}∆C_{m\left(m,k\right)}\approx C_{m\left(m,n\right)}+\left(N_{\left(m,n\right)}+\left(\alpha -1\right)N_{\left(m,1\right)}\right)$$
where N_(m,n)_ denotes the noise power coupled at C_m(m,n)_, α denotes the filter gain of the prefilter at RX[1], and ∆C_m(m,k)_ reflects the capacitance difference in C_m(m,k)_ and C_m(m,k−1)_. Assuming that the prefilter completely suppresses the N_(m,1)_ in C_m(m,1)_ (α = 0), the noise power at the C_m(m,n)_ node can be reduced to the difference in N_(m,n)_ and N_(m,1)_, both of which should have equal phase and amplitude for complete noise cancellation. This is another criterion for a differential sensing scheme.

In prior research [5], we assumed the coupled display noise as a global injection noise source, because the panel load of a rigid AMOLED is small. However, for a metal-mesh TSP with reduced TFE thickness in flexible AMOLED, the panel delays the TSP signal with an attenuation that causes a mismatch in the phase and amplitude of the coupled display noises. In addition, the method of selecting a reference C_m_s for the differential sensing scheme is yet to be reported.

Therefore, this study proposes a TSP system architecture that resolves the emerging display noise problem in a flexible AMOLED display. The proposed analog front end (AFE) combines discrete-time (D-T) filtering, multireference differential sensing (MRDS), and multi-capacitance driving (MCD) methods [13,14] to sufficiently attenuate the locally injected display noise. The AFE contains a single transconductance charge amplifier (CA) per ∆C_m_-sensing that acts as both a band-pass filter (BPF) and a ∆C_m_-to-voltage converter. The switched-capacitor (SC) correlated-double-sampler (CDS) executed the D-T BPF, and the following nondecimated programmable finite impulse response (FIR) filtering reduced the load of the AFE. The remainder of this paper is organized as follows. The display noise coupling in a flexible AMOLED, including the reconstruction method of differential sensing and the MRDS scheme, is reviewed in Section 2. The operation of the proposed AFE and the practical considerations in the circuit design are explained in Section 3. Thereafter, the measurement results of the AFE fabricated in a 350 nm CMOS process are presented in Section 4, and the conclusions of the current study, along with a brief summary, are presented in Section 5.

## 2. Differential Sensing Architecture

### 2.1. Circuit Model of Display Noise in Flexible AMOLED

The equivalent circuit model of the TSP, AMOLED pixel, and DDI illustrating the display noise insertion is presented in Figure 2a. During the horizontal-line refresh time (Scan[n]), the source drivers in the DDI generate RGB update signals. For simplicity, the update signal (V_DAC,R_[h]) for the h-th red pixel in the n-th horizontal line (P_R_[n,h]) is described. In addition, the V_DAC,R_[h] is generated from a digital-to-analog converter (DAC) and a unity gain amplifier, which was trapped on P_R_[n, h] through M_1_, M_2_, and C_S_. Simultaneously, the display noise (N_D_) propagated through the DDI parasitic capacitance (C_P_D_) to the cathode plane. For a rigid AMOLED with an encapsulation thickness of 100 μm and a C_P_TSP_ of tens of picofarad, most of the N_D_s leaked into the cathode plane. However, in a flexible AMOLED, the C_P_TSP_ of an RX electrode increases to approximately 1 nF. Consequently, such a large C_P_TSP_ introduces considerable display noise into the RX electrode that is directly coupled to the RX and AFE inputs. Moreover, a portion of the N_D_ propagates through an array of R_Cathode_ and C_P_TSP_, experiencing a phase delay and gain attenuation. The propagation of the display noises is depicted in Figure 2b, which implies that the phase delay is approximately equal to the time constant τ of the panel. As such, the τ between the RX[i] and RX[j] electrodes can be expressed by the distributed RC model as
(2)$$\;\;\;\;\;\;\;\;\;\;\;\;\;\;\;\;\;\;\;\;\;\;\;\;\;\tau =\frac{m\left(m-1\right)}{2}R_{Cathode}C_{P\_TSP}$$
where m denotes the number of RX electrodes between the RX[i] and RX[j] electrodes. In particular, the RX[i] electrode is directly affected by the display noise as it propagates into the RX[j]. The propagated display noise can be expressed as
(3)$$\;V_{N,n}=\gamma V_{DAC}\left(1-e^{-T_{sync}/\tau }\right)\overset{N_{BW}}{\underset{i=1}{\sum }}sin\left(2\pi \left(2i-1\right)f_{sync}\;\;\;\;+tan^{-1}\left(2\pi \left(2i-1\right)f_{sync}\tau \right)\right),\;\;\;\;\;$$
where V_DAC_ denotes the amplitude of V_DAC_[h], γ represents the amplitude reduction ratio owing to the capacitive divider of C_P_D_ and C_P_TSP_, and 2π(2N_BW_ − 1)f_sync_ indicates the noise bandwidth of the AFE. As the V_DAC_[h] is capacitively coupled, the display noise includes the derivative of V_DAC_[h] and its harmonic component. This feature signifies that the display noise in a flexible AMOLED is uncommon and cannot be eliminated by the conventional column-parallel differential sensing method.

Figure 3 shows the measured AFE readout signal for 16 × 34 C_m_s during 30 TSP frames where white image is displayed over whole screen. In the white color, the signal levels of V_DAC,R_[h], V_DAC,G_[h], and V_DAC,B_[h] are distributed in equilibrium, which generates regular display noise patterns to the TSP at a rate of f_sync_. Since the f_sync_ is not synchronized with TSP driving frequency, the fluctuating jitter can be viewed at the AFE readout signal when an induced noise level is significant. For the simplicity of analysis, the time-varying jitter (N_TV_) of readout signal is normalized to the readout signal of 4 mm conductive rod (∆S_4phi_), which is a minimum allowable touchable area for most commercialized mobile devices. The C_m(16,1)_–C_m(16,3)_ were located in the leftmost region of the TSP, and the other arrays of C_m(16,16)_–C_m(16,18)_ and C_m(16,32)_–C_m(16,34)_ were situated in the middle and rightmost regions of the TSP, respectively. Thus, the phase and amplitude of the coupled display noises were similar in each group of C_m_s, except for the capacitance offset. However, the phase and amplitude become uncorrelated as the distance between the C_m_s increases in the horizontal direction. Thus, the display noise is not differentially canceled for a flexible AMOLED display.

### 2.2. Capacitance Reconstruction in Multidriving, Differential Sensing Architecture

The C_m_ reconstruction from ∆C_m_ is illustrated in Figure 4. For simplicity, let us consider that C_m(1,1)_–C_m(4,4)_ are sensed by column-parallel differential sensing, and each TX electrode is simultaneously driven by 4-length orthogonal sequences. In this case, the AFE sensed the encoded capacitance difference ∆C_m_e(1,1)_–∆C_m_e(1,3)_. Thus, the sensed readout signal (=3) is one less than the number of RX electrodes. We assumed that a finger is touched between the C_m(3,2)_ and C_m(3,3)_, wherein the fringing field (∆C) contacts the finger and decreases the C_m(3,2)_ and C_m(3,3)_. At this instant, the ∆C_m_e(1,2)_ becomes zero, and positive ∆C_m_e(1,1)_ and negative ∆C_m_e(1,3)_ are generated. After A/D conversion, the decoder demodulated ∆C_m_e(1,1)_~∆C_m_e (1,3)_ into ∆C_m(1,1)_ ~∆C_m(4,3)_. Simultaneously, an integral basis or the reference C_m_s (C_ref(1)_–C_ref(4)_) was derived to execute the integral of ∆C_m(1,1)_~∆C_m(4,3)_ based on C_ref(1)_–C_ref(4)_. However, a large noise disturbance upon C_ref(1)_–C_ref(4)_ is undesirable, because it acts as a global noise.

In addition to the required noise suppression on C_ref(1)_–C_ref(4)_, we carefully considered the selection of the reference RX location, as discussed later. The root–mean–square jitters (N_TV_RMS_) during 100 TSP frames are presented in Figure 5a, which were measured following the same approach as that in Figure 3. In contrast, the RMS noise measured following a differential-sensing manner is depicted in Figure 5b. Here, the 16-C_m_ on the RX[0] electrode was used as an integral basis for reconstruction.

As shown in Figure 5a, the noise power in the single-ended measurement was approximately equipotential throughout the TSP area. Furthermore, the variation between N_TV_RMS_MAX_ and N_TV_RMS_MIN_ was 0.015. On the other hand, the noise power of the differential sensing increased at specific locations. The N_TV_RMS_AVG_ was 0.021 for the C_m(1,1)_–C_m(16,17)_, which was located on the left-hand side of the RX[17] electrode. This is approximately half of that for single-ended sensing. In contrast, the average and peak RMS noises (N_TV_RMS_AVG_, N_TV_RMS_MAX_) for C_m(1,18)_–C_m(16,34)_ increased to 0.046 and 0.065, respectively, because the signal path of the C_m(1,18)_–C_m(16,34)_ experienced a larger delay. This resulted in an amplified mismatch of the coupled display noises between the reference C_m_ and sensing C_m_. Therefore, the location of reference C_m_ should be carefully selected.

### 2.3. Proposed MRDS Scheme

The phase mismatch of the display noises was minimized by selecting the C_m_s on an RX electrode located at the center of the TSP as an integral basis. For simplicity, we assumed that a TSP with 40 RX electrodes was used, α in Equation (1) was zero, and RX[20] was considered the reference electrode. Upon subtracting the noises coupled at RX[n] and RX[20] electrodes for n < 20, a phase lag was generated in the noise-difference signal by adding a negative phase, and vice versa for n > 20. This solution could reduce the overall phase delay and yield the maximum m in Equation (2) as 20. The reconstructed C_m_ with a single reference RX electrode at the TSP center can be expressed as
(4)$$\;\;\;C_{m\_rct\left(m,n\right)}=\left\{\begin{array}{l} -\overset{N/2}{\underset{k=n}{\sum }}∆C_{m\left(m,k\right)}+C_{m\left(m,N/2\right)}\;\;\;\;\;\;\;\;\;\;\;,\;n\leq \frac{N}{2}\\ \overset{N}{\underset{k=N/2+1}{\sum }}∆C_{m\left(m,k\right)}+C_{m\left(m,N/2\right)}\;\;\;\;\;\;\;\;\;,\;n>\frac{N}{2} \end{array}\right.$$
where N denotes the number of the RX electrodes. More importantly, the noise-cancellation effectiveness can be improved by adopting a multi-RX reference electrode. In a practical consideration, the location of the multi-RX reference electrodes was selected such that it minimized the overall phase lag and the lead of noise-difference signals. Accordingly, a reference RX electrode was placed at the center of the left-half plane of the TSP, whereas the other was placed at the center of the right-half plane. Therefore, the reconstructed C_m_ of the MRDS scheme with two reference RX electrodes can be obtained as follows.


(5)
$$\;\;\;C_{m\_rct\left(m,n\right)}=\left\{\begin{array}{l} -\sum_{k=n}^{\frac{N}{4}}∆C_{m\left(m,k\right)}+C_{m\left(m,\frac{N}{4}\right)}\;\;\;\;\;\;\;\;\;\;\;,\;n<\frac{N}{4}\\ \sum_{k=\frac{N}{4}+1}^{\frac{N}{2}}∆C_{m\left(m,k\right)}+C_{m\left(m,\frac{N}{4}\right)}\;\;\;\;\;\;\;\;\;\;\;,\frac{N}{2}>n\geq \frac{N}{4}\\ \sum_{k=\frac{N}{2}+1}^{\frac{3}{4}N}∆C_{m\left(m,k\right)}+C_{m\left(m,\frac{3N}{4}\right)}\;\;\;\;\;\;\;\;\;\;\;,\frac{3N}{4}>n\geq \frac{N}{2}\\ \sum_{k=\frac{3}{4}N+1}^{N}∆C_{m\left(m,k\right)}+C_{m\left(m,\frac{3N}{4}\right)}\;\;\;\;\;\;\;\;\;\;\;,\frac{3N}{4}>n\geq \frac{N}{4} \end{array}\right.$$


The feasibility of the MRDS scheme was validated by evaluating the display noise cancellation using several time constants (τ) in Equation (2) and a horizontal update rate (w_sync_ = 2πf_SYNC_). In addition, we assumed that there are 40 RX electrodes, and the differential display noise powers were scaled with that of the single-ended sensing. In Figure 6, three cases are plotted versus the product of w_sync_ and τ: selection of the reference RX electrode as RX[0] in a conventional method, the RX[20] centered on the TSP, and two RX[10] and RX[30] in the proposed MRDS scheme. The simulated result revealed that the noise with reference RX[0] becomes greater than that obtained from the single-ended sensing method, if w_sync_τ > 0.002. For reference RX[20], the noise power decreased in comparison with the reference RX[0]. However, as w_sync_τ increased, the differential cancellation effect was reduced to an insignificant level. In the case of using two references (RX[10] and RX[30]), the noise power was maintained below 0.4 until w_sync_τ reached a value of 0.009.

In a practical design, the w_sync_τ does not exceed 0.009. Accordingly, let us consider that a TSP with a C_P_TSP_ of 1 nF and R_cathode_ of 5 Ω, and f_sync_ of 300 kHz can be used. Notably, this is the worst possible case for a large-sized flexible AMOLED TSP, such as a tablet or note PC applications, wherein the w_sync_τ reaches 0.0094. As such, the overall SNR was lower than that of the reference RX. In this design, the number of RX in the MRDS scheme was set to 2 to achieve a reasonable tradeoff. For a TSP with C_P_TSP_ of 500 pF, R_cathode_ of 10 Ω, and f_sync_ of 76.8 kHz, the w_sync_τ was calculated as 2.4 × 10^−3^, wherein the noise powers were approximately 0.5 and 0.13 for reference RX[0] and the two references of RX[10] and RX[30], respectively. Therefore, the MRDS scheme was effective if the noise power at the reference RX electrode exceeded 0.13.

## 3. Differential Front-End Architecture

### 3.1. Block Description

The proposed MRDS front-end architecture is illustrated in Figure 7. It comprises a TSP driver, reconfigurable CAs, D-T sampling CDS, A/D converter, and decoding and reconstructing logics. In addition, it sensed the TSP with 16 TX and 34 RX electrodes in a flexible AMOLED. Overall, the entire TSP operation cycle was categorized into two phases: C_ref_- and ∆C_m_-sensing. In the ∆C_m_-sensing phase, the TSP driver simultaneously stimulated four TX electrodes using the MCD method. Specifically, the MCD sequences of C_ref_ were reconfigured such that its length was four times longer than that of ∆C_m_. As compared with ∆C_m_, it could spread the in-band (IB) noise interference at the TSP driving frequency (f_DRV_) by 6 dB. At the RX electrodes, the CA executed a differential-to-single-ended (D-S) conversion along with a continuous-time (C-T) BPF for out-of-band interference. After ∆C_m_-sensing, the CA operation was configured as a single-ended sensing circuit, handling the single-ended C_ref_. The CA was followed by the D-T CDS, which band-passed the samples and down-converted the desired signals at f_DRV_ to DC. As the desired signal was already bandpass-filtered at the CA, the distortion from the noise folding caused by the sampling in the CDS posed a minor concern. In addition, the CDS outputs were converted to digital values by the time-interleaving operation of a successive approximation register (SAR) ADC via a multiplexer. As the desired signal was already filtered at both the CA and CDS stages, the SAR ADC executed only the A/D conversion. Thus, this architecture is power- and area-efficient in comparison to the ∆Ʃ ADC-based AFE [10] that requires high-frequency operating clocks for decimation and should be placed at every RX electrode. The ADC readout data for C_ref_ (C_ref_e(1)_) and ∆C_m_ (∆C_m_e(1,1)_–∆C_m_e(4,33)_) were processed in the same approach described earlier (Figure 4).

The proposed front-end architecture posed several advantages. First, the C-T BPF and D-S conversion can be performed with a single capacitive-feedback CA, unlike the fully or pseudo-differential sensing circuits [7,15] which require several capacitive-feedback amplifiers at multiple RX stages for implementing differential sensing and noise filtering. This allows the utilization of more RX channels than the TX channels in the TSP and the corresponding front-end blocks to be integrated into the designed chip. Thus, it can increase the allowable sensing time per ∆C_m_ to improve the signal-to-noise ratio (SNR). Second, the MRDS scheme is an extremely flexible method in which the location and number of reference RX can be selected according to the R-C load of the TSP. Accordingly, various TSP sizes can be supported by increasing the power dissipation only during the C_ref_-sensing phase.

However, the proposed front-end architecture poses certain limitations—one of them is insufficient noise filtering that causes residual noise interferences in the signal path, which is a simple first-order C-T BPF. In addition, the CDS features first-order moving-average BPF. To enhance the noise suppression, the FIR filtering was applied to the output of the C_ref_ reconstruction at the TSP frame rate (f_TSP_). Thus, the effect of the MRDS scheme was evaluated considering the worst possible scenario in which a high-definition (HD) flexible AMOLED display with 1280 × 720 resolution and 60 Hz update rate was used.

### 3.2. Analog Front-End Circuit

The configurations of the proposed CA and timing diagram of 3-to-2 multiplexer for both ∆C_m_- and C_ref_-sensing are presented in Figure 8 and Figure 9, respectively. As illustrated in Figure 9, the entire TSP operation cycle can be divided into ∆C_m_-sensing (T_DIFF_) and C_ref_-sensing (T_REF_), digital signal processing (DSP) such as interpolation and coordinate filtering, and idle time. During T_DIFF_, the 3-to-2 multiplexer connected the two adjacent RX electrodes to the (+) and (−) inputs of the CA. In particular, the T_DIFF_ comprised two detection phases for high S_ODD_ and S_REF_[17:1] (Figure 8a), Rx[3] and Rx[2] electrodes were connected to (+) and (−) inputs of the 1st CA (V_IN1+_, V_IN1-_) to output encoded capacitance difference (C_rd_) placed in Rx[3] and Rx[2] electrodes whereas Rx[2] and Rx[1] electrodes were connected in case of high S_EVEN_. Consequently, the number of AFEs blocks was half of that of RX electrodes. More specifically, four of the 16 TX electrodes were simultaneously driven by 4-length encoded Baker (4-LBK) sequences [16]. The 4-LBK exhibited perfect orthogonality and increased the sensing period by four times as compared to that of the sequential driving, thereby increasing the SNR by 6 dB. Owing to the differential sensing, the base capacitance that remained unchanged by finger touch was differentially negated as well. Thus, the cancellation technique is not required for *∆**C_m_*-sensing. Additionally, a D-S conversion was performed in the CA, comprising a single-ended output op-amp and an array of capacitors and resistors in the feedback path (C_IN_, R_IN_) and V_IN+_ (C_IP_, R_IP_). For positive C_m_-sensing passing through V_DRV_ and from V_IN+_ to V_CA_ O_, the R_IP_ and C_IP_ were placed in parallel positions between the V_IN+_ and the ground of the system (V_G_). Therefore, the positive C_m_-signal transfer function with the application of the 4-LBK encoding can be expressed as
(6)$$\begin{array}{c} \;G_{CA+}\left(s\right)=\frac{sV_{DRV}\sum_{i=1}^{4}a_{i}C_{m\left(i,N\right)}R_{IP}}{(1+s\sum_{i=1}^{4}a_{i}C_{m\left(i,N\right)}R_{I})\left(1+sC_{IP}R_{IP}\right)+s\sum_{i=1}^{4}a_{i}C_{m\left(i,N\right)}R_{IP}}\;\\ \times \frac{(1+s\sum_{i=1}^{4}a_{i}C_{m\left(i,N\right)}R_{I})\left(1+sC_{IN}R_{IN}\right)+s\sum_{i=1}^{4}a_{i}C_{m\left(i,N\right)}R_{IN}}{\left(1+s\sum_{i=1}^{4}a_{i}C_{m\left(i,N\right)}R_{I}\right)\left(1+sC_{IN}R_{IN}\right)}\\ ≅\;\frac{sV_{DRV}\sum_{i=1}^{4}a_{k}C_{m\left(i,N\right)}R_{IP}}{\left(1+sC_{IN}R_{IN}\right)\left(1+s\sum_{i=1}^{4}a_{i}C_{m\left(i,N+1\right)}R_{I}\right)} \end{array}$$
where a_i_ denotes the coefficient of 4-LBK, and C_m(i,N)_ represents the mutual capacitance at the cross-section of the TX[i] and RX[N] electrodes. Upon matching the sizes of the R_IP_ and C_IP_ with those of the R_IN_ and C_IN_, the transfer function became equal to that of the negative C_m_-sensing path, except for the sign passing through V_DRV_ from V_IN-_ to V_CA_O_. Thus, the differential C_m_ (∆C_m_) sensing can be achieved.

The input resistor R_I_ was placed at both the inputs of the CA and generated a high-frequency pole 1/(R_I_Ʃa_i_C_m(i,N+1)_) to provide low-pass filtering ability onto the CA controlled by the 4-bit. During CDS sampling, this prevented the noise-folding from interferences of frequencies higher than f_DRV_. In previous studies, the AFE contained several op-amps with multiple stages to create high-order BPF and C-V conversion. In this study, the proposed CA contains only a single op-amp to implement the first-order BPF and D-S conversion. Therefore, the SNR per AFE power consumption and area were enhanced. This solution increased the number of RX channels as compared to the TX channels in the AFE, where a sufficient sensing period can be assigned to ∆C_m_-sensing. Moreover, the op-amp in the CA is a two-stage class AB output amplifier, which supports the dynamic currents from C_m_. Based on the post-layout simulation, its unity gain bandwidth was marginally set at 30 times (15 MHz) than that of the usable f_DRV_ with 70-dB DC gain.

During T_REF_ for C_ref_-sensing, the proposed CA was tuned to a single-ended sensing circuit [17] in which V_IN+_ was shorted to V_G_. As the time-varying noise on C_ref_ can act as a global-injection noise source, a more stringent noise immunity was imposed on the C_ref_-sensing. Herein, we used the 16-LBK that simultaneously stimulated all the TX electrodes and increased its in-band spreading gain to 12 dB. However, the 16-LBK is limited by a poor dynamic range at the CA, which yields a code summation equal to 4. In addition, a majority of the summed signal was occupied by the base capacitance, which did not alter upon a finger touch and was required to be removed. The C_IP_ at V_IN+_ during ∆C_m_-sensing was reused to negate the C_ref_ offset, where the upper side of the C_IP_ was connected to V_IN-,_ and the lower side was driven by a negative V_DRV_. Therefore, the base capacitance of the C_ref_ could be canceled without any additional circuits.

The BPF of reconfigurable CA was followed by D-T bandpass sampling of the CDS, which is illustrated using a simplified noise model in Figure 10. In the CDS, the CA output V_CA,O_ of f_DRV_ was down-converted to DC, which reflects the difference between the V_CDS,OP_ and V_CDS,ON_. Moreover, the ∆C_m_-proportional charge was tracked onto C_INT,CDS_ with the dynamic switching of S_IP_ and S_IN_. On the contrary, the interference was attenuated by windowed integrating prefilter [18], which operated based on the integrate-and-hold (I/H) function of the S_IP_, S_IN_, and D-T CDS. The gain (G_CDS_) was determined by the capacitance ratio of C_IN,CDS_/C_INT_ and the G_CDS_ in a passband region, which was not PVT-sensitive. Furthermore, the noise components can be categorized into four independent sources: external noises coupled onto the TSP $(\bar{V^{2}{}_{ext}}$), input-referred noise of the CA $\left(\bar{V^{2}{}_{CA}}\right)$, thermal noise of the on-resistances $\left(\bar{V^{2}{}_{SW,on}}\right)$ of S_IP_ and S_IN_, and CDS noise ($\bar{V^{2}{}_{CDS}}$). In particular, the $\bar{V^{2}{}_{ext}}$ and $\bar{V^{2}{}_{CA}}$ were reduced by G_CA_ in (5) and the non-inverting gain (1 + G_CA_) from V_IN+_ to V_CA,O_. Thereafter, all the noise sources, including $\bar{V^{2}{}_{SW,on}}$ and $\bar{V^{2}{}_{CDS}}$, were sampled during the integration (I) phase. The duty cycle (γ) of the I-phase should increase for selective filtering, because the filter notches of the windowed integration occurred at multiples of $2f_{DRV}/\gamma$. Moreover, the non-overlapping of the S_IP_ and S_IN_ is required to prevent charge sharing between the positive and negative D-T integrations in the CDS.

After sinc-windowed integration of the I/H circuit, the interference was further suppressed by two steps: D-T analog FIR (AFIR) filtering in which the coefficients of +1 and −1 were expressed by the alternating closings of S_IP_ and S_IN_, and a pseudorandom spread of MCD sequences at the rate f_DRV_/N_AFIR_, where N_AFIR_ denotes the number of closings of S_IP_ and S_IN_. During the ∆C_m_-sensing phase, the code length (N_LBK_) of the 4-LBK sequence was 4, which decimated the scan rate of ∆C_m_ down to f_DRV_/(4N_AFIR_). As 16 TX electrodes were considered, the T_DIFF_ (Figure 9) is equal to 2f_DRV_/N_AFIR_. Notwithstanding, a more stringent noise suppression is required for C_ref_ to facilitate the noise requirement of the AFE, because the SNR for reconstructed *C_m_* is eventually limited to that of C_ref_, assuming that the common noises are canceled by differential sensing. The SNR enhancement method for C_ref_ using 4-tap digital FIR (DFIR) filtering in the *z*-domain is portrayed in Figure 11, wherein the mutual capacitances on the RX[j] and RX[k] electrodes were assumed as C_ref_s and decoded into C_ref,j(1)_–C_ref,j(16)_ and C_ref,k(1)_–C_ref,k(16)_, respectively. In addition, Ref_k_[n] and Ref_j_[n] describe the n-th frame of the C_ref,j(1)_–C_ref,j(16)_ and C_ref,k(1)_–C_ref,k(16)_. In addition, the latest four Ref_k_[n:n − 3] and Ref_j_[n:n − 3] denote the inputs of the 4-tap DFIR filter, which were weighted by b_1_–b_4_ and summed to provide C_ref,j(1F)_–C_ref,j(16F)_ and C_ref,k(1F)_–C_ref,k(16F)_. In this case, the filtered C_ref_ was not decimated as it reflected the time-varying C_m_ of the TSP.

The overall filter response of the AFE is presented in Figure 12 and can be expressed as
(7)$$H\left(f\right)=N_{LBK}N_{AFIR}\frac{C_{IN,CDS}}{C_{INT,CDS}}sinc\left(\frac{\gamma f}{2f_{DRV}}\right)e^{-j\pi \frac{\gamma f}{2f_{DRV}}}H_{D-T}\left(f\right)$$
where the discrete-time transfer function H_D-T_(f) is stated as
(8)$$\;\;\;H_{D-T}\left(f\right)=\sum_{i=0}^{N_{AFIR}-1}{\left.z^{-i}\right|}_{z=e^{\frac{j\pi f}{f_{DRV}}}}\sum_{i=0}^{N_{LBK}-1}{\left.a_{LBK_{i}}z^{-i}\right|}_{z=e^{\frac{j\pi f}{N_{AFIR}f_{DRV}}}}H_{DFIR}\left(f\right)$$
where *a_LBK_i_* denotes the i-th MCD sequence, and the reconfigurable DFIR transfer function H_DFIR_(f) can be written as
(9)$$\;\;\;\;\;\;H_{DFIR}\left(f\right)=\overset{N_{DFIR}-1}{\underset{i=0}{\sum }}{\left.b_{i}z^{-i}\right|}_{z=e^{\frac{j\pi f}{\left(N_{AFIR}N_{LBK}f_{DRV}\right)}}}$$

In Equation (7), the four components can be distinguished as gain, sinc-windowed prefilter, cumulative integration in the analog domain, and FIR filter in the digital domain. The aggregation of these transfer functions for C_ref_ and *∆*C_m_s sensing is depicted in Figure 13. It is simulated with an *N_AFIR_* of 30, γ of 1, and a DFIR length of 6. The I/H operation significantly influences the transfer function roll-off around the 2*f_DRV_.* Consequently, the in-band depth was enhanced by 9 dB, implying that the achievable SNR was limited by the transfer function for C_ref_.

### 3.3. TSP Driver

Although the SNR for the TSP readout system can be enhanced via the MCD scheme for a large number of TX electrodes, the radiated electromagnetic interference (EMI) emission from the TSP electrodes increases and interferes with the frequency band of wireless communications at the scale of several hundreds of megahertz. In addition, several metal traces were placed parallel to a number of TX pads in the TSP driver, which are typically long. Moreover, wide top-metal interconnected wires in the chip can generate EMI to additional on-chip components when driving the TSP. As depicted in Figure 14, the proposed excitation power controlled the TSP driver integrated onto the pad. Specifically, it comprised two output buffers (D1 and D2), control logic, transmission gate switch (TG), and pull-down nMOS transistors. The designed pad could be configured as either TSP driving or sensing by enabling Drv_en and Sen_en, respectively. Therefore, a wide range of TSP channel configurations can be supported without any additional circuitry.

For the general-purpose I/O configuration, controlling the Pd_en transforms the pin into an open-drain driver with an external pull-up resistor. In addition, small series resistors (R1-R4) are inserted in the signal path of each configuration to prevent a short circuit at the output of the pad. For high Drv_en and low Str_en, the stimulation power of the excitation signal could be controlled by Str_ctr<3:0>, depending on the application. However, the selection of the D1 as the output buffer can reduce the TX power without disturbing the frequency bands of the digital communication standard for cellular phones. The rising and declining slopes of the control signal for the TX excitation were controlled by the pre-driver, which controlled the width/length ratio of the transistors in the pre-driver as well. Overall, the integration minimized the EMI power produced in the signal path from the Drv[n] to the pad output.

## 4. Measurement Results

The measurement environment is illustrated in Figure 15a, which portrays the prototype IC mounted on a flexible connector module occupying a chip area of 4.8 mm^2^, which includes proposed CA, CDS, 3-to-2 Mux, SAR ADC and digital logic. The AFE was tested with a 6.7-inch flexible AMOLED with 16 TX × 34 RX TSP sensors. It consists of 0.5 mm bendable polyester (PET) film as a cover glass, polarizer, metal-mesh TSP TX and RX electrodes in the 3^rd^ layer (L3), 15 μm thin-film encapsulation glass to protect the thin film transistor (TFT) and RGB pixel from oxidation and moisture permeability, and the TFT circuit in the 1^st^ layer (L1). The display update rate was set to 60 Hz with a 1280 × 720 resolution HD display, exhibiting an f_SYNC_ of 76.2 kHz. In addition, a microscope photograph of the TSP TX and RX electrodes (in the L3) were present in Figure 15a and described in Figure 15b, which contained a wide array of fine-line diamond grid metal-mesh patterns with numerous openings, where the RGB pixels (in L1) were placed in a pixel pitch of 50 μm. A mutual capacitance node (C_m,p_) was generated by cutting the metal lines at the border of the TX and RX electrodes. A number of C_m,p_s form single mutual capacitance (C_m(1TX/1RX)_) between the TX and RX electrodes. Additionally, the TSP patterns were opened within a TX or RX electrode. This fractal structure increased the fringing field between the finger and TSP electrodes in presence of a finger touch. The measured RX C_P_TSP_ of the 6.7-in TSP is illustrated in Figure 16. The C_P_TSP_ was measured with the f_DRV_ of 10 kHz to remove a signal reduction effect from large TSP electrode resistance and C_P_TSP_. The display driving was turned off only for identifying the C_P_TSP_ without any display noise disturbance. The measured RX C_P_TSP_ ranged from 470–549 pF. As C_P_TSP_ is generated between the TSP electrode and cathode plane, the minimum C_P_TSP_ occurred at both ends of the TSP owing to a lack of coupling. The time-varying measured outputs in the differential and single-ended tuned CA in cases of no display image and updating the zebra-pattern over the display are presented in Figure 17a,b, respectively.

To prevent the AFE saturation in harsh noisy conditions, one-third of the dynamic range was used for the capacitance signal, and the single-ended CA output swing was 1.06-V with an f_DRV_ of 340 kHz. Moreover, the coded-summed signal swing of ∆C_m_s was 510 mV. In this case, the noise amplitude is reduced from 0.88 to 0.28 V, which was further attenuated by the band-pass FIR filtering of the CDS and the spreading of the 4-LBK. The measured power spectral density (PSD) depicted in Figure 17b is further illustrated in Figure 18. As the pixel-charging signal was a square wave during Scan[1:H], the display noise contained several odd-harmonic components of f_sync_. According to the measurement, the noise amplitude was reduced by 10.2 dB in comparison to that of single-ended sensing, and the noise-immunity performance converged to that of the C_ref_s.

The effectiveness of the MRDS scheme was validated by measuring the time-varying raw capacitances using a conventional differential sensing method (Figure 19a) and the MRDS scheme (Figure 19b). These measurements were conducted over a 100-TSP frame and RMS-averaged over the entire TSP.

Accordingly, the RX[17] was selected for the conventional differential scheme, whereas the RX[9] and RX[26] electrodes were selected for the MRDS scheme. In addition, the TSP must be driven on the exact instant in the display frame for quantitative analysis. Thus, the initiation of the TSP driving is synchronized with the vertical porch of the DDI at all instances, where two TSP frames form a single display frame of 60 Hz.

This indicated that the C_m(m,n)_ in a specific location is affected by the same display noise for every measurement in all cases. In both the measurements, the noise amplitude gradually increased with the distance between the reference and sensing electrodes owing to the imbalance of the coupled display noises. However, the phase delay was effectively reduced in the MRDS scheme, producing a maximum *m* of 8. With a signal variation of approximately 400 upon the touch of a 4 mm-diameter conductive rod, the maximum noise amplitude located at the rightmost area of the TSP reduced from 97 to 44, thereby increasing the SNR by 6.84 dB. The SNR plot displaying the external noise insertion from a radiation plate under the panel is presented in Figure 20, which ranged from DC to 800 kHz with a 1-kHz step. To verify the C_ref_-SNR improvement with the 16-LBK and DFIR schemes, the SNR was measured for both the single-ended sensing and MRDS schemes without the display noise coupling. As the noises from a radiation plate were uniformly distributed without experiencing a delay, the overall SNR was equal to the SNR of C_ref_ and increased it by 7 dB in comparison to single-ended sensing.

As displayed in Figure 21, the feasibility of the excitation power control in the TX pad was evaluated. For this measurement, the near-field EMI scanner was placed on the chip and was connected to a spectrum analyzer. Moreover, the EMI power was evaluated at the frequency band of the global system for mobile communications (GSM)-750/850, ranging from 747.2–893.8 MHz and requiring the suppression of blocker signals by 70 dBm [19]. Upon enabling the output buffer of D1 in Figure 15, the averaged EMI level was reduced by 9 dB and resulted in an average of −73.4 dBm /Hz in the two measurements. The SNR versus TSP scan rate is plotted in Figure 22. The SNR is defined in [20,21] and measured in the same manner. As the scan time increases (the scan rate decreases), the number of excitations per each measuring of ∆C_m_ or C_ref_ increases, proportionally. It can be seen that as the scan time is doubled, the SNR increased by approximately 3 dB. This is because the known signal for mutual capacitance (∆C_m_ or C_ref_) adds linearly whereas the noise portion adds as the square root of the sum of the squares. The proposed AFE consumed 17.6 mW, and the total power breakdown of the AFE is charted in Figure 23.

Furthermore, the performance summary of the proposed method was compared with those reported in prior research adopting differential sensing, as listed in Table 1. The comparative analysis revealed that the proposed AFE required the lowest power consumption per number of C_m_ channels, and the SNR directly relied on the panel structure and performance of the AFE. In a flexible AMOLED containing a metal-mesh sensor, TFE, and cathode plane, the strongest fringing field of C_m_ is transmitted into the T_P_TSP_ owing to the reduced thickness (15 μm) of the TFE. Therefore, the SNR of 50.5 dB can be further enhanced by using a standalone metal-mesh TSP sensor without the TFE and cathode layer or rigid AMOLED with increased TFE thickness. The prototype achieved comparable energy along with an area efficiency figure of merit of the touch readout system.

## 5. Conclusions

This study proposes a noise-immune touch readout AFE for a flexible AMOLED display, wherein a multireference differential sensing scheme was applied to minimize the mismatch of the coupled noise interferences in a bandwidth-limited TSP. The proposed charge amplifier can be reconfigured as either pseudodifferential or single-ended sensing, and the bandpass noise filtering was applied to prevent both external noise and noise-folding from the correlated double sampling stage. In addition, a panel excitation circuit was integrated onto the pad to minimize the radiated EMI, which can be used for sensing purposes as well. Thereafter, the proposed AFE was implemented in a 350 nm CMOS process and tested with a 6.7 in HD flexible AMOLED panel with 16 TX and 34 RX electrodes. The developed AFE achieved a 50.5 dB SNR with 17.6 mW power consumption. In conclusion, the proposed device achieved a high SNR and low operation power in addition to the improved noise immunity against severe display noise interference of flexible AMOLED displays.

## Figures and Tables

**Figure 1 micromachines-13-00942-f001:**
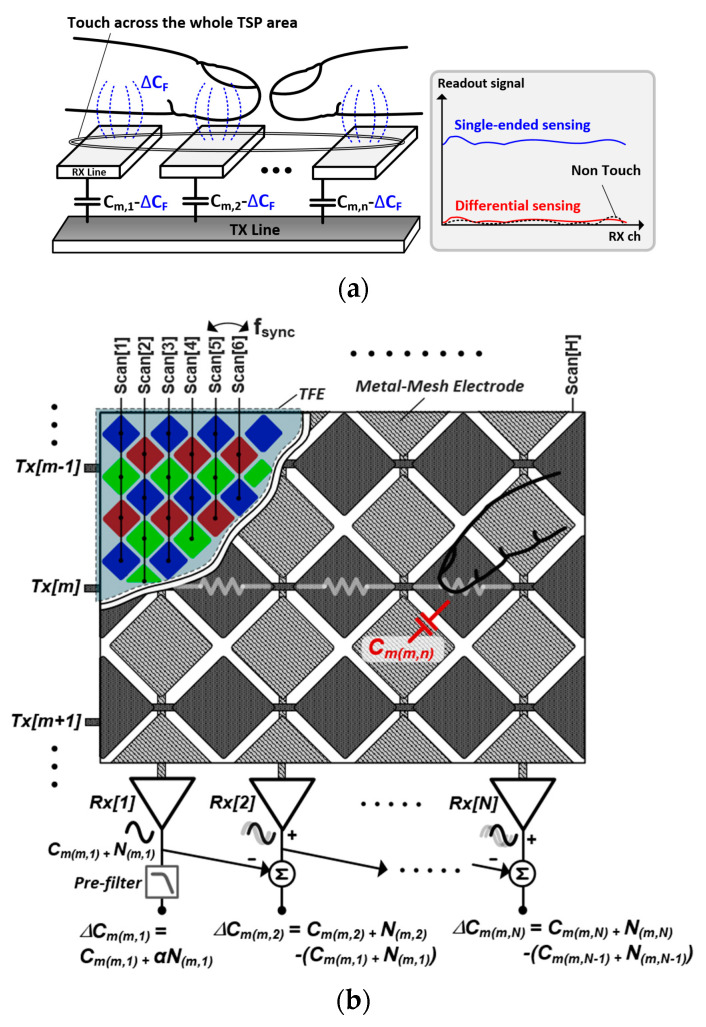
(**a**) Multi-touch user scenario showing missing touches of differential sensing and (**b**) conceptual diagram of proposed differential sensing scheme and stack-up structure of flexible AMOLED display.

**Figure 2 micromachines-13-00942-f002:**
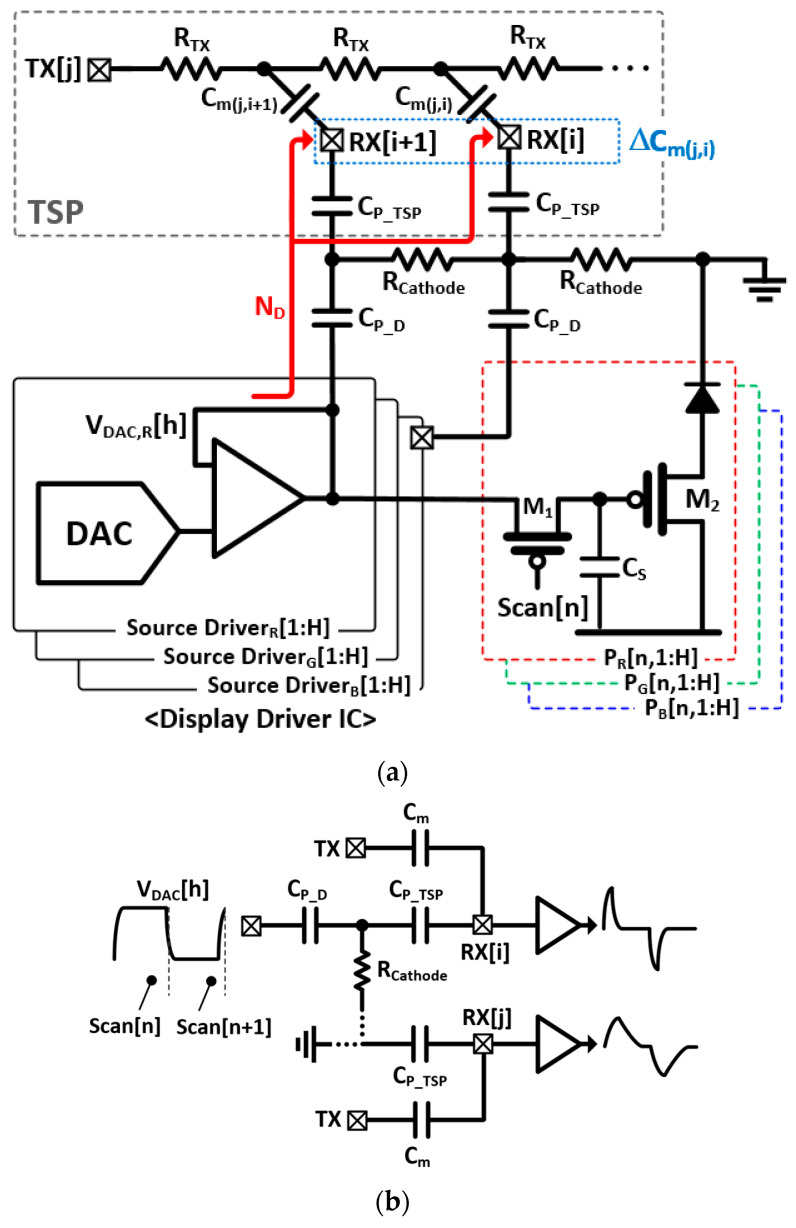
(**a**) Equivalent circuit model of DDI, TSP, and AMOLED pixels and (**b**) simplified model illustrating display noise injection.

**Figure 3 micromachines-13-00942-f003:**
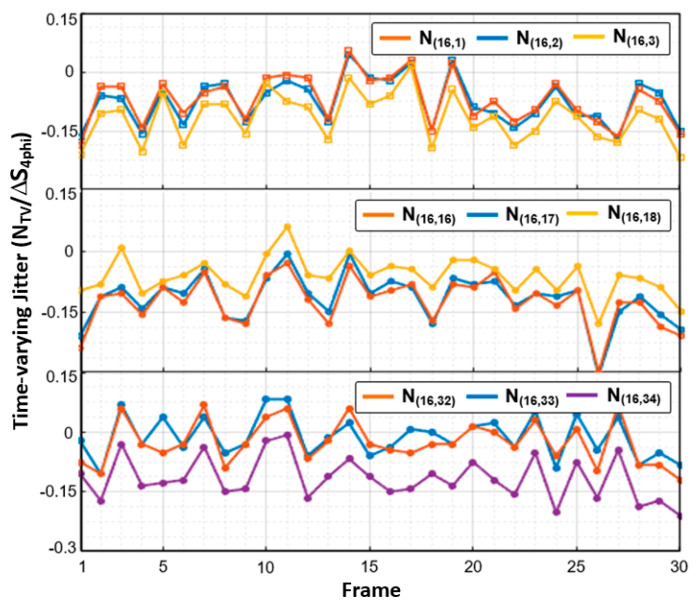
Measured time-varying jitter values at the leftmost, middle, and rightmost area of the TSP for 16 × 34 flexible AMOLED display.

**Figure 4 micromachines-13-00942-f004:**
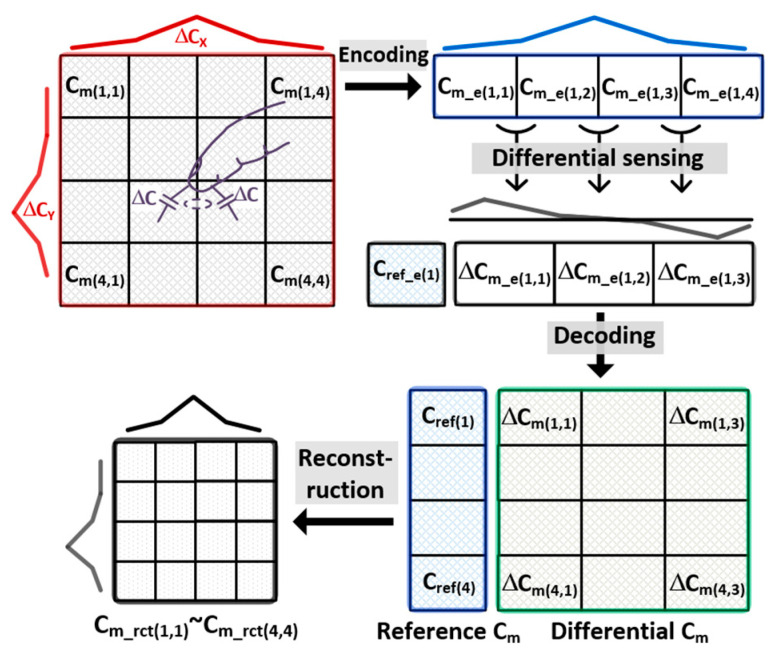
Reconstruction of single-ended capacitances.

**Figure 5 micromachines-13-00942-f005:**
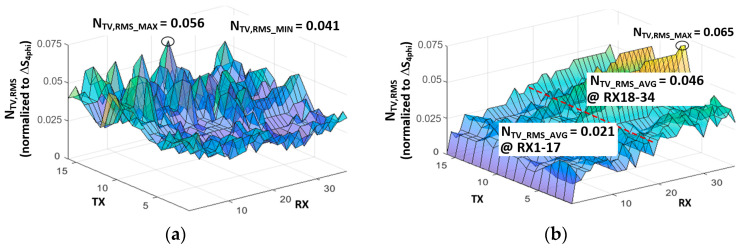
RMS noise values measured in (**a**) single-ended and (**b**) differential-sensing manners.

**Figure 6 micromachines-13-00942-f006:**
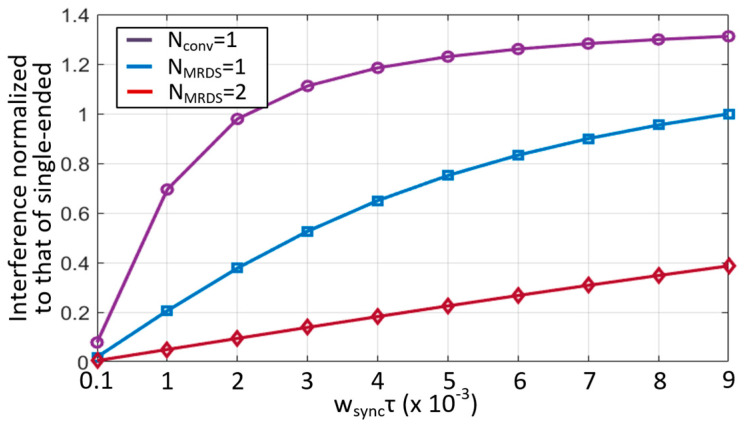
Comparison of interference cancellation effect in a conventional differential sensing and proposed MRDS scheme.

**Figure 7 micromachines-13-00942-f007:**
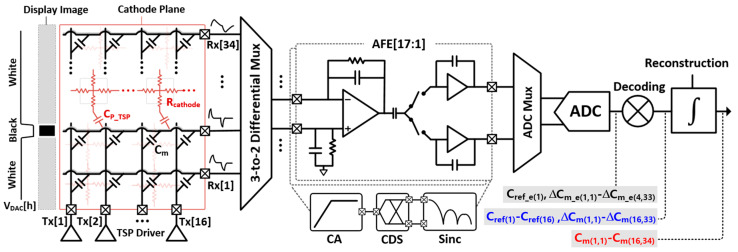
Proposed MRDS front-end architecture.

**Figure 8 micromachines-13-00942-f008:**
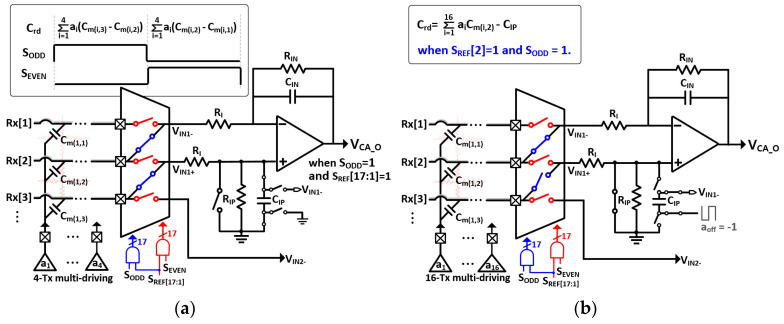
Proposed reconfigurable CA for both (**a**) sensing of ∆C_m_s and (**b**) C_ref_s.

**Figure 9 micromachines-13-00942-f009:**
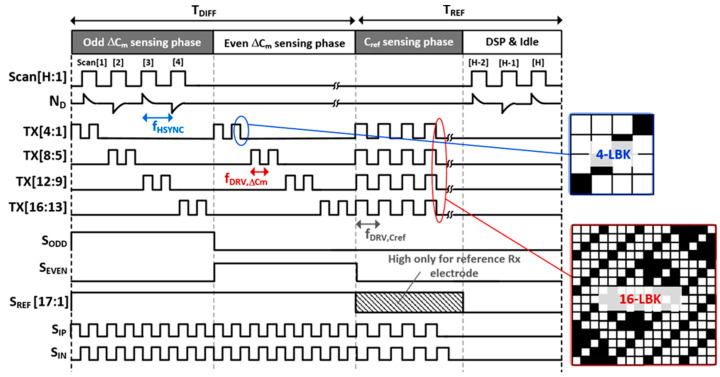
Timing diagram for illustrating the operation of proposed AFE with reconfigurable MCD sequences.

**Figure 10 micromachines-13-00942-f010:**
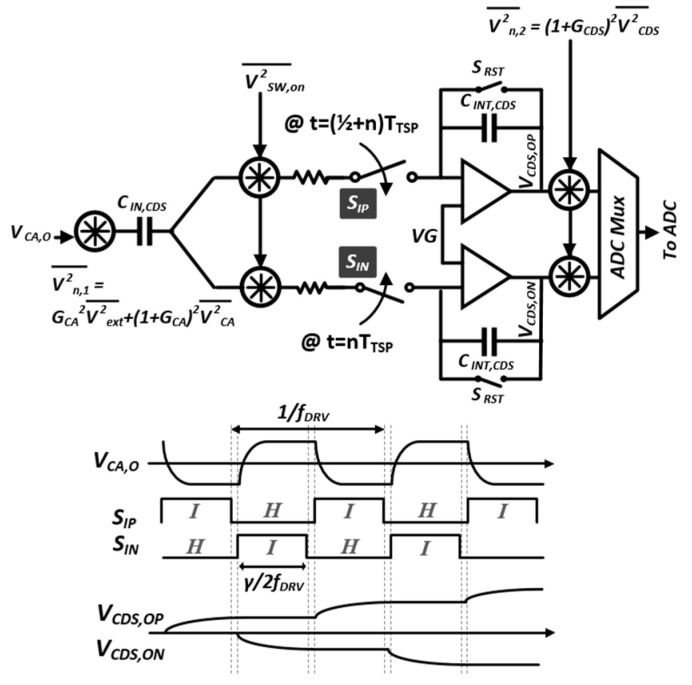
Schematic and timing diagram of the CDS with equivalent noise model.

**Figure 11 micromachines-13-00942-f011:**
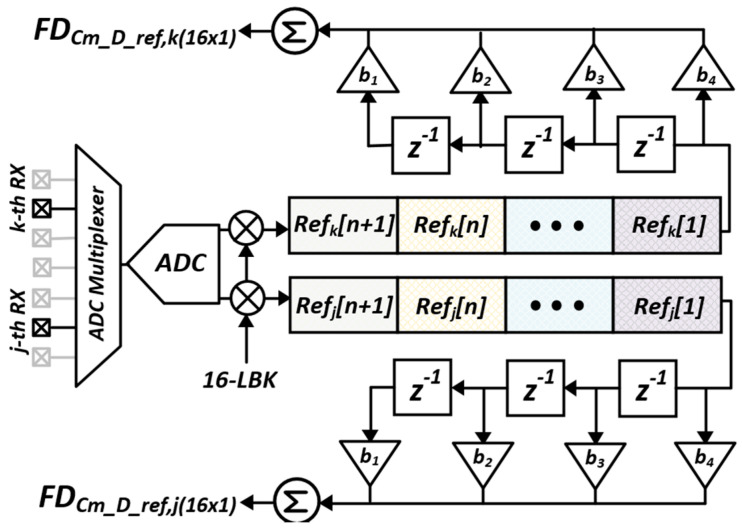
DFIR filtering scheme to improve SNR for C_ref_.

**Figure 12 micromachines-13-00942-f012:**
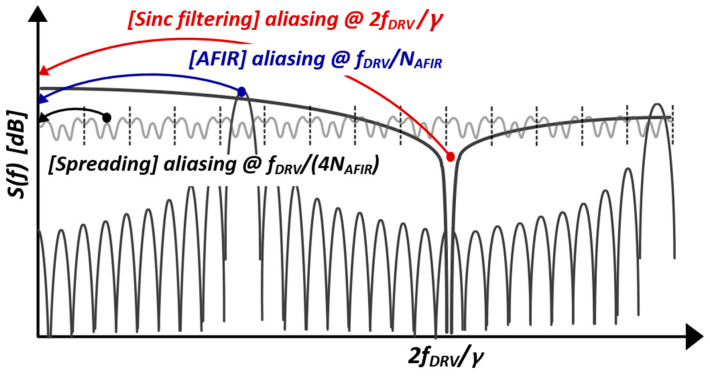
Overall cascaded transfer function of AFE.

**Figure 13 micromachines-13-00942-f013:**
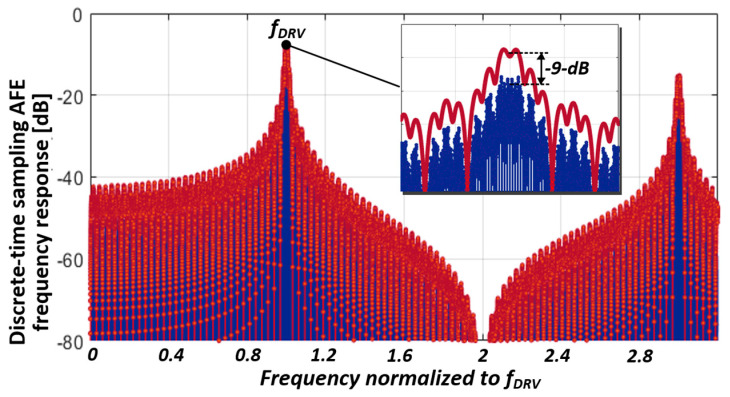
Simulated aggregated transfer function of AFE.

**Figure 14 micromachines-13-00942-f014:**
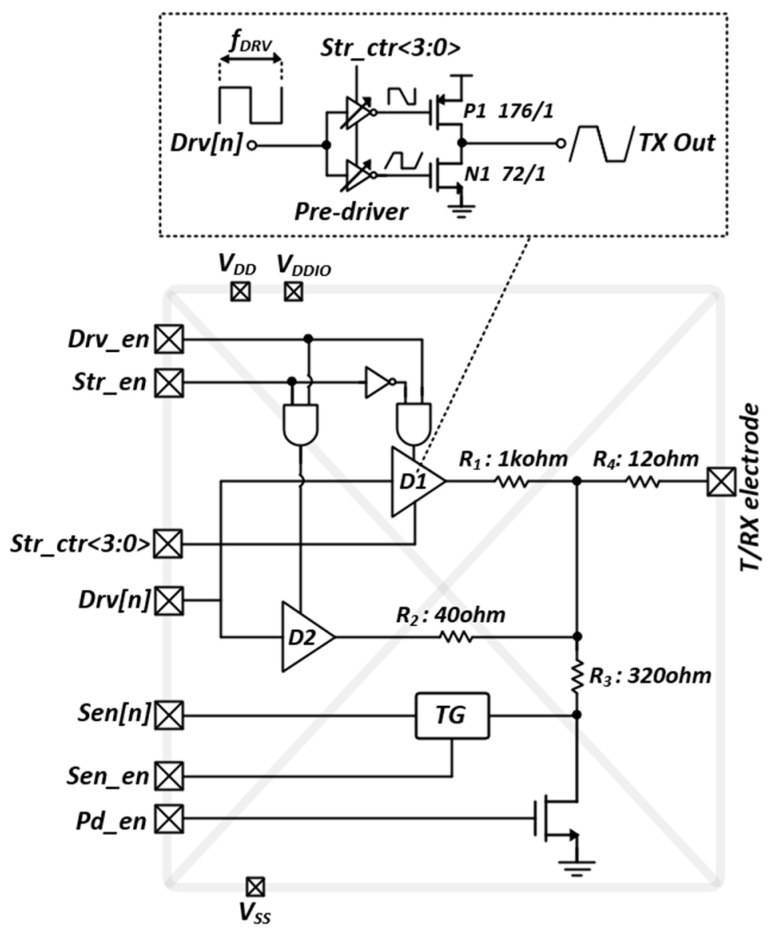
Schematic of reconfigurable pad circuit for both driving and sensing of TSP.

**Figure 15 micromachines-13-00942-f015:**
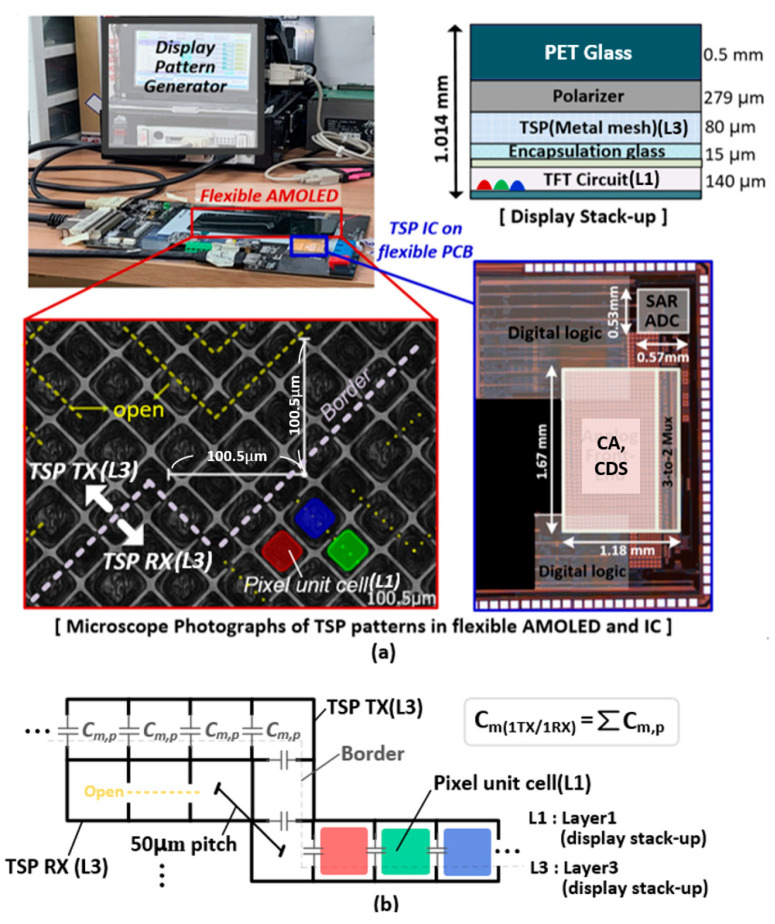
(**a**) Measurement environment and micrographs of TSP and IC, and (**b**) overall diagram of metal-mesh TSP with display pixel.

**Figure 16 micromachines-13-00942-f016:**
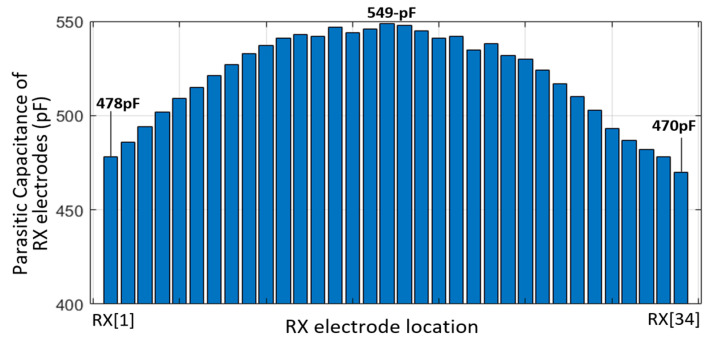
Measured parasitic capacitance of RX electrodes.

**Figure 17 micromachines-13-00942-f017:**
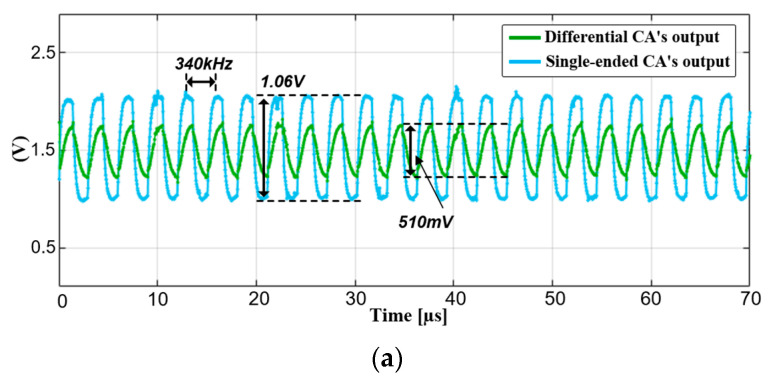
Measured time-varying outputs of differential and single-ended configured CA (**a**) without, and (**b**) with noisy display-update.

**Figure 18 micromachines-13-00942-f018:**
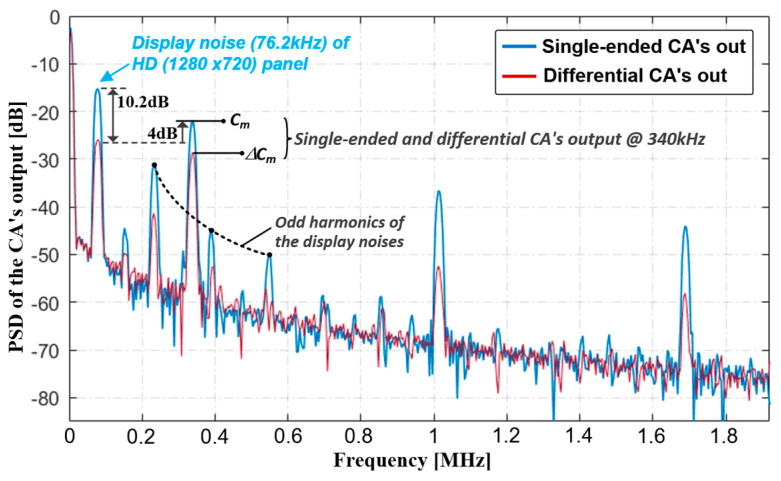
PSDs of CA output with display noise interferences.

**Figure 19 micromachines-13-00942-f019:**
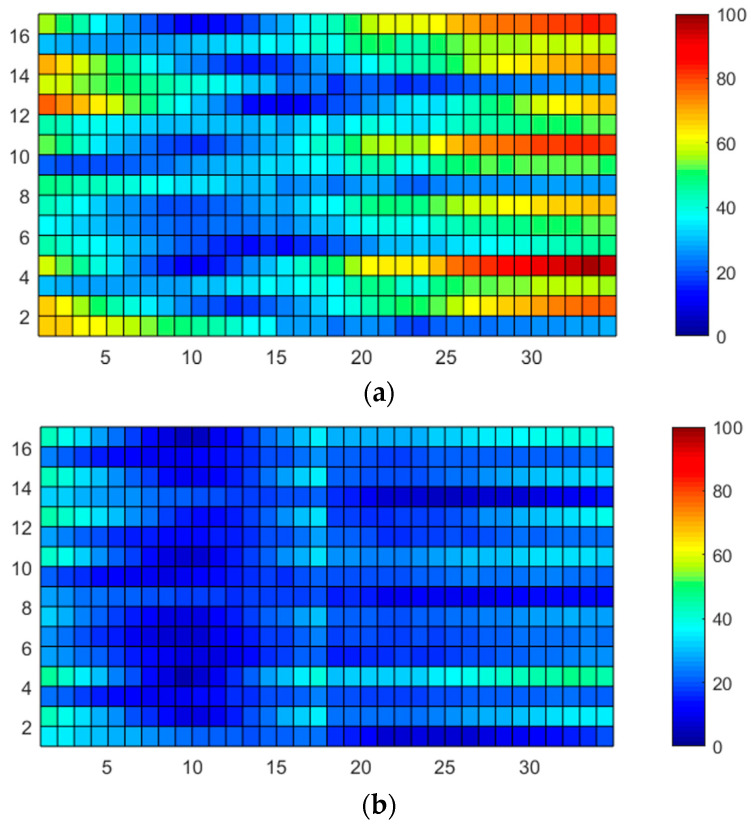
RMS noise-jitter values of TSP raw data for (**a**) single RX-line reference and (**b**) MRDS scheme.

**Figure 20 micromachines-13-00942-f020:**
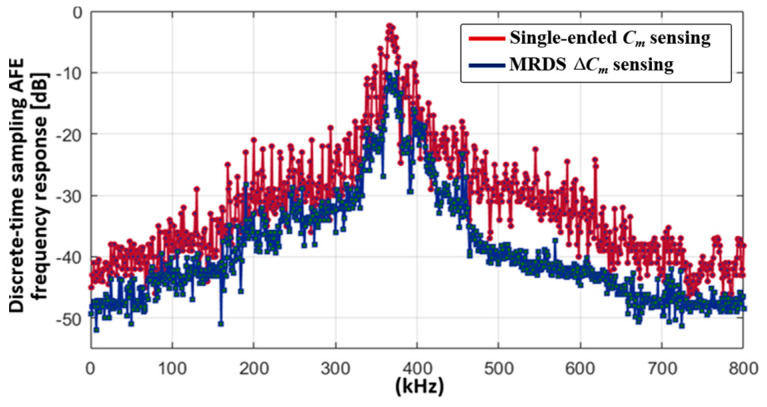
Measured noise suppression ratio from DC to 800 kHz for single-ended and MRDS-applied configurations.

**Figure 21 micromachines-13-00942-f021:**
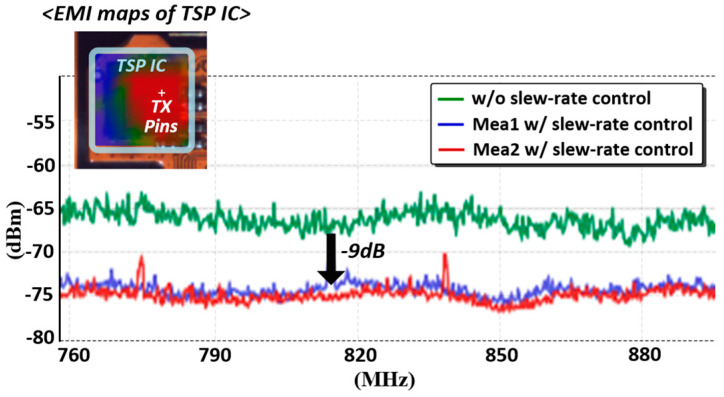
PSD of radiated TSP driving signals in the GSM frequency band.

**Figure 22 micromachines-13-00942-f022:**
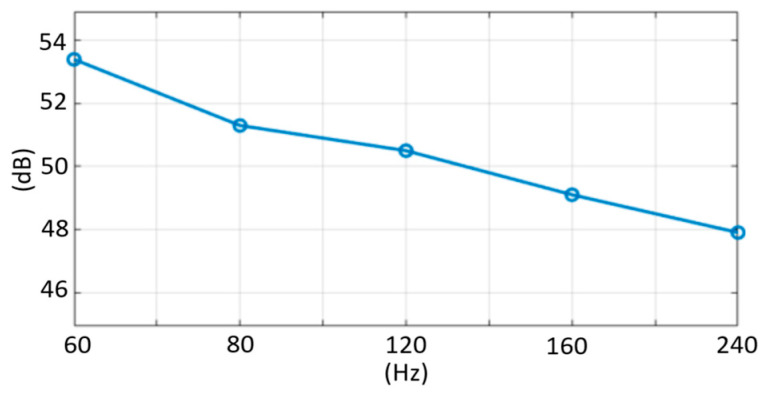
Measured SNR versus scan rate.

**Figure 23 micromachines-13-00942-f023:**
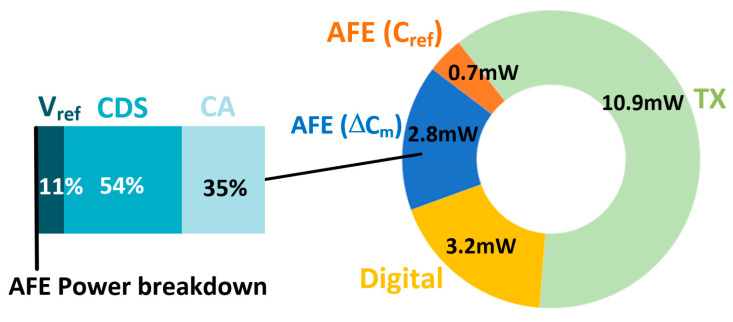
Power breakdown of designed AFE.

**Table 1 micromachines-13-00942-t001:** Performance summary and Comparison to Previous Works.

	Present Study		[7]	[8]	[10]	[21]	[22]
Process	0.35 μm (AFE) 0.13 μm (Digital)		0.18 μm	0.35 μm	90 nm	0.35 μm	0.18 μm
Electrodes	TX 16 × RX 34		TX 36 × RX 64	TX 12 × RX 8	TX 28 × RX 16	TX 29 × RX 53	TX 78 × RX 138
Scan rate	200 Hz		85~385 Hz	200 Hz	200 Hz	140 Hz	240 Hz
Power Consumption (PC)	17.6 mW		67.7 mW	6.26 mW	24.6 mW	19 mW	559.9 mW
PC/Electrode	22.8 μW		29.3 μW	65.2 μW	54.9 μW	12.3 μW	52 μW
Chip area	4.8 mm^2^		36 mm^2^	2.2 mm^2^	15.9 mm^2^	20 mm^2^	71.2 mm^2^
SNR	Flexible AMOLED	50.5 dB	54 dB	60 dB	60 dB	12.6 dB	56.6 dB
	Rigid AMOLED	59 dB					
Panel Size/ TSP Type	6.7″ Metal Mesh Flexible AMOLED		12.2″ ITO LCD	4.3″ ITO	5″ On-cell ITO AMOLED	13.3″ ITO	32″ Metal Mesh LCD
FoM ^(1)^ (pJ/step)	2.31		2.11	0.717	4.36	399	23.2
TSP Capacitance Sensing method	Multi-reference reconstruction		Column Parallel	Column Parallel	Column Parallel	Column Parallel	Column Parallel
In-band spreading sequence	16-lengh Barker		36-length HM ^(2)^	-	-	-	255-length MLS ^(3)^

(1) FoM = Chip area × Power/(SNR · # of node · scan rate) (2) HM = Hadamard matrix (HM) multi-driving sequence, (3) MLS = Maximum length multi-driving sequence.

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
