# Peer review of "Robust Touch Screen Readout System to Display Noise Using Multireference Differential Sensing Scheme for Flexible AMOLED Display"

_micromachines, 2022, doi:10.3390/mi13060942_

Round 1

Reviewer 1 Report

I suggest that this work be accepted following minimal revisions.

1) The manuscript's introduction is too lengthy; please make it shorter.

2) A comparative study in the form of a table must be included in the updated manuscript.

Reviewer 2 Report

I have some comments.

  1. The originality of this paper is not clear. The major feature of the proposed design should be compared with the published works of the authors.
  2. The proposed design is implemented in 130-nm process. However, the power supply voltage of 3V is too high. This point should be explained clearly, such as device/circuit reliability problem ?
  3. The used power supply voltage is high (3V). However, the achieved SNR is very poor (50.5 dB) as in Table I. This issue should be discussed.
  4. There's figure caption error in Figure 6 ?
  5. This paper can be considered as a complete work with test chip measurements.

Round 2

Reviewer 2 Report

The quality of this revised version has been improved. I may consider

this version suitable for publication.